# Multiple Timescale Dynamic Analysis of Functionally-Impairing Mutations in Human Ileal Bile Acid-Binding Protein

**DOI:** 10.3390/ijms231911346

**Published:** 2022-09-26

**Authors:** Gergő Horváth, Bence Balterer, András Micsonai, József Kardos, Orsolya Toke

**Affiliations:** 1NMR Research Laboratory, Centre for Structural Science, Research Centre for Natural Sciences, 2 Magyar Tudósok Körútja, H-1117 Budapest, Hungary; 2ELTE NAP Neuroimmunology Research Group, Department of Biochemistry, Institute of Biology, ELTE Eötvös Loránd University, Pázmány Péter sétány 1/C, H-1117 Budapest, Hungary

**Keywords:** bile acids, enterohepatic circulation, intracellular lipid-binding proteins, ligand binding, NMR spectroscopy, positive cooperativity, site-selectivity, protein dynamics

## Abstract

Human ileal bile acid-binding protein (hI-BABP) has a key role in the enterohepatic circulation of bile salts. Its two internal binding sites exhibit positive cooperativity accompanied by a site-selectivity of glycocholate (GCA) and glycochenodeoxycholate (GCDA), the two most abundant bile salts in humans. To improve our understanding of the role of dynamics in ligand binding, we introduced functionally impairing single-residue mutations at two key regions of the protein and subjected the mutants to NMR relaxation analysis and MD simulations. According to our results, mutation in both the vicinity of the C/D (Q51A) and the G/H (Q99A) turns results in a redistribution of motional freedom in *apo* hI-BABP. Mutation Q51A, deteriorating the site-selectivity of GCA and GCDA, results in the channeling of ms fluctuations into faster motions in the binding pocket hampering the realization of key side chain interactions. Mutation Q99A, abolishing positive binding cooperativity for GCDA, leaves ms motions in the C-terminal half unchanged but by decoupling *β*D from a dynamic cluster of the N-terminal half displays an increased flexibility in the vicinity of site 1. MD simulations of the variants indicate structural differences in the portal region and mutation-induced changes in dynamics, which depend on the protonation state of histidines. A dynamic coupling between the EFGH portal, the C/D-region, and the helical cap is evidenced highlighting the interplay of structural and dynamic effects in bile salt recognition in hI-BABP.

## 1. Introduction

Bile salts are amphipathic molecules (Figure 1A) synthesized from cholesterol in the liver, which in the small intestine facilitate the absorption of lipid-like compounds [1,2,3]. In addition, by interacting with receptors [4,5,6,7] and activating various signalling pathways [8,9], they support the regulation of gene expression and cellular function. This includes the regulation of their own synthesis and the maintenance of cholesterol homeostasis and a contribution to triglyceride, energy, and glucose metabolism. With a role in a diverse set of activation processes, bile acid transport and bile acid-controlled signalling pathways are important targets for the treatment of metabolic disorders and diseases of the gastrointestinal tract [10,11,12]. During digestion, bile salts are efficiently recycled via a process termed enterohepatic circulation [1] involving multiple membrane transport events and the transcellular trafficking of bile salts. Human ileal bile acid binding protein (hI-BABP), a member of the family of intracellular lipid-binding proteins (iLBPs) [13,14], expressed in the epithelial cells of the distal small intestine, has a key role in the cytoplasmic transport of bile salts in enterocytes.

Similar to other iLBPs [13], the binding cavity of BABPs is enclosed by two antiparallel five-stranded β-sheets and a helix-loop-helix motif at the top (Figure 1B) [15,16,17,18,19]. Despite the common topology, BABPs exhibit unique properties in the family of iLBPs [20]. Unlike fatty acid binding proteins (FABPs) and cellular retinol binding proteins (CRBPs), BABPs have been shown to have two internal binding sites displaying positive binding cooperativity in both the human ileal [21,22] and the chicken liver forms [23]. In human I-BABP, the energetic communication between the binding sites has been shown to be governed by the hydroxylation pattern of the bound bile salts [24] and as shown for glycocholate (GCA), its contribution can exceed the intrinsic affinity of the binding sites [21]. In addition, human I-BABP has been found to show a high degree of site-selectivity in its interactions with GCA and glycochenodeoxycholate (GCDA), the two most abundant bile salts in humans [25]. Specifically, while in homotypic complexes di- and trihydroxy bile salts occupy both binding sites, in the heterotypic complex of hI-BABP/GCDA/GCA, they displace each other and selectively occupy site 1 and site 2, respectively (Figure 1B,C).

NMR relaxation measurements show evidence of the presence of μs-ms conformational fluctuations in FABPs, CRBPs, and BABPs, and indicate that internal motions have a major role in ligand-protein recognition in iLBPs [26]. In FABPs and CRBPs, the helical segments exhibiting considerable disorder in the *apo* state together with the proximate C/D-turn are thought to form a dynamic portal region mediating access to the enclosed binding cavity [27,28,29]. Fluorescence study of a triple mutant of murine adipocyte FABP supports this hypothesis by showing that an enlargement of the gap between the C/D-turn and the helical cap markedly increases the accessibility of the ligands to the cavity [30]. Investigation of a helix-less variant of rat I-FABP shows the abolishment of a rate-limiting kinetic step in binding [31], whereas a helix-less variant of rabbit I-BABP demonstrates a strong coupling between binding and folding [32]. Similar to FABPs, stopped-flow kinetic analysis of full-length human I-BABP indicates a rate-limiting conformational transition on the ms timescale before the first binding step [33].

Relaxation dispersion (*R_ex_*) NMR measurements at room temperature show evidence of internal motions in *apo* human I-BABP matching the timescale of the unimolecular kinetic step preceding binding and show that bile salt binding results in the ceasing of slow motions [34]. The extension of *R_ex_* measurements to lower temperatures reveals the presence of two different exchange processes on the ms timescale affecting two distinct protein regions; a faster cluster comprising the EFGHIJ *β*-strands of the C-terminal half and a slower cluster involving segments of the BC-region and part of the helical cap in the N-terminal half [35]. An enthalpy-entropy compensation is indicated for both clusters between the ground and a sparsely populated higher energy state characteristic of order-disorder transitions. Supporting this notion, a joint analysis of *R_ex_* measurements and NMR thermal melting data suggests a structural and thermodynamic connection between ms timescale conformational fluctuations at and below room temperature and thermal unfolding (60 °C) of human I-BABP [36]. In agreement with the NMR relaxation data, MD simulations of *apo* human I-BABP show evidence of correlated motions involving regions in both the N- and the C-terminal half (helical cap, *β*E-*β*F, and bottom of the *β*-barrel) [36].

Conformational exchange on a similar timescale is known to also exist in chicken liver BABP [37,38]. However, unlike in hI-BABP, slow motions in cL-BABP are confined to the C-terminal half. The protonation state of histidines is thought to have a key role in the conformational exchange dynamics in both cL-BABP [17,37] and hI-BABP [39]. As we have shown recently by pH dependent NMR measurements, conformational fluctuations in the human analogue are most pronounced near the *pK_a_* of the histidines [39]. Cessation of slow motion upon either bile salt binding or lowering the pH in the *apo* form, as revealed by *R_ex_* measurements, indicates similarities in the dynamic behavior of the complexed and the histidine-protonated unligated forms of the protein. Regarding the structural characteristics of the *R_ex_*-detected higher energy state, the correlation between *R_ex_*-derived ^15^N chemical shift differences between the ground and the excited state (|Δδ|) as well as the chemical shift difference between *apo* and *holo* hI-BABP in the C-terminal half suggests that the higher energy state possesses a more open *holo*-like conformation in the EFGH-region in the absence of ligands [35]. The connection between thermal melting and the *R_ex_*-detected global conformational fluctuations further suggests that the higher energy state carries partially unfolded characteristics with an opening in the EFGH-region [36]. In addition to the abolishment of slow fluctuations below the *pK_a_* of H98, changes in |Δδ| at pH = 6.3 and above suggest that similarly to bile salt binding, histidine protonation at H98 stabilizes the more open conformation of the EFGH *β*-strands [39]. Of note, the opening of the EFGH-region is accompanied by a closure of the helical cap and the gap between the helical and the C/D-turn upon both ligand binding [19] and as revealed by MD simulations, in *apo* hI-BABP as well [36]. Based on the available structural and dynamic data, a conformational selection mechanism of ligand binding is hypothesized in both the chicken liver and the human ileal analogues [19,37], the details of which are not yet fully clear. In particular, the role of the dynamic cluster in the N-terminal half in the human analogue needs a better understanding. Likewise, the contribution of local and long-range effects in binding cooperativity and site selectivity has not been fully explored.

To gain more insight into the role of flexibility in human I-BABP function and improve our understanding of the regulation of ligand binding, we have introduced two single-residue mutations into two key regions of the protein and subjected the mutants to a multiple timescale motional analysis by NMR relaxation measurements and MD simulations in the *apo* state. Each of the two mutations has previously been shown to alter the characteristics of ligand binding [22,33] allowing us to relate residue level dynamic differences obtained from NMR to mutation-induced changes in the thermodynamics and kinetics of bile salt binding. Specifically, residue Q51 is located in the proximity of the C/D-turn (Figure 1B) and its mutation to alanine has been shown to abolish the site-preference of GCDA and GCA without affecting positive binding cooperativity [22]. The other mutated residue, Q99, is located at the beginning of *β*H and its mutation to alanine results in faster binding accompanied by the loss of positive cooperativity for GCDA with no observed change in the site preference of di- and trihydroxy bile salts [22,33]. One possibility is that the observed differences in the macroscopic characteristics of binding arise from local enthalpic effects. Alternatively, mutation-induced changes in site-selectivity and binding cooperativity can result from allostery. As we show here, structural perturbations introduced by the mutation at these two pliable sites are propagated to distant regions in *apo* hI-BABP. Redistribution of motional freedom resulting from each of the two functionally-impairing mutations raises the possibility of the modification of the binding properties of human I-BABP by modulating the flexibility of specific protein segments in the protein.

## 2. Results

### 2.1. Overall Structure and Stability Studied by CD Spectroscopy

To compare the overall structure and thermodynamic stability of wild-type and mutant human I-BABP variants, CD spectroscopy experiments were carried out as a function of temperature. At room temperature, the three variants exhibited nearly identical far-UV CD spectra characteristic of proteins containing antiparallel *β*-sheet as main structural feature. Increasing the temperature at 5 °C steps, CD spectra were collected up to 85 °C (Figure 2A and Appendix A). All the variants showed a cooperative and fully reversible thermal unfolding with a melting point close to 60 °C (Appendix A). The secondary structure composition of the variants as a function of temperature was estimated from the CD spectra using the BeStSel method [40,41] (Figure 2B). The expectable error of the secondary structure estimation is around 5 percent in the absolute values [40,41], however, because of the similar spectral shape and amplitude of the WT and mutant proteins, it shows more accurately the structural changes in comparisons. These results show that the mutants are fully folded and their overall structure and stability is similar to the WT protein. Unexpectedly, the *α*-helices, which correspond to the N-terminal “cap” of hI-BABP, loosen up at lower temperatures, well before the global unfolding transition. This phenomenon is well expressed for wild-type and Q51A hI-BABP with midpoint temperatures of 53.8 ± 0.7 and 53.0 ± 0.8 °C (fitted by a two-state unfolding model), respectively, and possibly has implications for the role of the helical cap in the process of ligand binding in these variants. Intriguingly, in the case of Q99A, the melting of *α*-helices (56.3 ± 0.8 °C) is rather linked to the overall unfolding of the molecule (Figure 2B). To further investigate the overall stability of the three variants, thermal denaturation experiments with continuous heating were carried out. The melting was followed by the CD signal at 200 nm wavelength, which is mainly contributed by the antiparallel *β*-sheets of the molecule, and the observed curves were fitted by a two-state unfolding model (Figure 2C). Measurements were carried out at pH 5.8, 6.3, and 8.0, matching the conditions, where transition between a closed and a more open protein state has previously been investigated by NMR [39]. Although at pH 6.3 the melting temperature of WT was only 1 °C and 1.6 °C higher than that of Q99A and Q51A, respectively, these differences are significant and observable in a wider pH range with somewhat larger effect at pH 8.0. (Appendix A and Appendix A). The unfolding enthalpy was calculated from the fitted curve for each variant. At pH = 5.8 and 6.3, values of *ΔH_unfold_* remained similar within the error for each variant, whereas at pH = 8.0, Q99A exhibited a slightly smaller enthalpy of unfolding in comparison to wild-type and Q51A. For WT, the unfolding enthalpy has ranged between 80.9 ± 1.7 kcal/mol (pH 6.3) and 89.4 ± 2.0 kcal/mol (pH 8.0) (Appendix A).

### 2.2. Chemical Shift Perturbation Mapping

To explore the effect of mutations on the protein backbone and the overall structure of the protein, backbone amide chemical shifts of the mutated human I-BABP variants have been compared with that of the wild-type. Combined *Δδ*_1*HN,*15*N*_ chemical shift differences [42] have been calculated on a per-residue basis and are depicted in Figure 3. In Q51A, values of *Δδ*_1*HN,*15*N*_ exceeding the average extend from the site of mutation to the neighbouring strands including the entire length of *β*B (I36-V40) and *β*D (T58-T64). Somewhat smaller but considerable chemical shift changes are observed in the EF-region (S69, M74-T78, T82), in a segment of *β*G (V92, F94), and in the C-terminal half of *α*-I (F17, L20, G22). Moreover, via H-bond interactions, the change in the chemical environment upon the mutation is transmitted to E7 of *β*A and two proximate positively charged residues in its vicinity (K124, R125).

In Q99A, the largest chemical shift differences from wild type are predominantly observed in the C-terminal *β*-sheet, primarily in *β*H in the vicinity of the mutation (N96-E102) as well as in the neighboring *β*G (L90-F94) and *β*I (T00, V111, T113, G115) strands, and the beginning of *β*J (T118, Y119) indicating that mutation-induced structural changes are transmitted by strand-strand interactions. Additionally, subtle perturbations in the chemical environment upon the mutation are indicated by values of *Δδ*_1*HN,*15*N*_ slightly exceeding the average in the EF-region (I71-G75) and at some key elements of the N-terminal half. One affected residue is F17, a bulky aromatic residue of *α*-I involved in multiple van der Waals interactions stabilizing the helical segment and by an interaction with Y119, establishing a connection between the helical cap and the C-terminal back of the *β*-barrel. Additional residues in the N-terminal half with significant (^1^HN, ^15^N) chemical shift change upon the mutation include N13 in the *β*A/α-I linker in the vicinity of F17 and Y119, as well as N33 in the *α*-II/*β*B linker.

### 2.3. Fast Protein Motions by NMR

Because of their strong effect on entropy, motions on the picosecond to nanosecond time scale have high biological importance [43]. To characterize the effect of the mutations on the ps-ns dynamics of the protein backbone, ^15^N T_1_, T_2_ relaxation and steady-state {^1^H}-^15^N NOE measurements were performed on Q51A and Q99A human I-BABP, subjected to model-free analysis [43,44,45,46,47], and compared to previously reported results on the WT protein [35]. The average values of longitudinal and transverse relaxation rates and {^1^H}-^15^N NOE for the investigated human I-BABP variants together with the mean values of the generalized order parameters (*S*^2^) characterizing the spatial restriction of the ^15^N-^1^H bond vector, are given in Table 1. Similar to our previous observation on the WT form, the majority of residues could be fit by the single-time scale (*τ_m_*, correlation time for macromolecular tumbling) model-free formalism using *S*^2^ alone (model 1) in each of the variants. Additionally, alike in WT, a subset of residues indicated the contribution of microsecond to millisecond conformational exchange to transverse relaxation (*R_ex,app_*, model 3), and in Q99A a non-overlapping, in Q51A a partially overlapping group of residues required the inclusion of an effective correlation time (*τ_e_*) for internal motion on the ps timescale as well (model 2 with *R_ex,app_* = 0, and model 4 with *R_ex,app_* > 0). In comparison to the WT protein, significantly less residues indicated the presence of conformational exchange in both mutants. Specifically, while in the wild-type protein 30, in Q51A 11 and in Q99A 14 residues required the inclusion of a *R_ex,app_* term. The number of residues requiring a term for ps motions were 14, 22, and 17 in WT, Q51A, and Q99A, respectively. The two amino acid positions suggesting the presence of superimposed µs-ms and ps motions in Q51A were T60 (*β*D) and G75 (E/F-turn). Motional parameters along the protein sequence obtained from the Modelfree-analysis for the three variants are listed in Appendix A.

Differences in the contribution of slow motions between the investigated variants involved (i) *α*-I, where as opposed to WT and Q51A, contribution from *R_ex,app_* in Q99A was not observed and (ii) *α*-II and *β*F, where the *R_ex,app_* contribution was less prevalent in both mutants than in WT. In the G-H region, a similar contribution of µs-ms motions was observed in each of the three variants. Regarding the ps timescale motions, *β*B and the linker between *α*-II and *β*B have been found to be substantially more flexible in both mutants than in the wild-type form. While the C/D- and E/F-turn regions have shown a similar contribution of *τ_e_* in each variant, two distant turn regions (F/G and H/I) exhibit enhanced flexibility in Q51A.

Similar to wild-type hI-BABP, the generalized order parameter in Q51A and Q99A averages around *S*^2^ = 0.91 and 0.92, respectively, suggesting no major change in the overall ps-ns flexibility of the protein backbone upon the mutation. However, a more detailed comparison of the three variants reveals subtle differences. This is depicted in Figure 4, where differences in *S*^2^ between wild type human I-BABP and the investigated mutants are shown on a per-residue basis. Specifically, in Q51A, an increase in the flexibility of the upper segments of the neighboring *β*B and *β*D is observed with enhanced ps-ns fluctuations being transmitted further to *β*A and the E/F-turn. In the case of Q99A, an increase of fast motions is observed in *β*E, the vicinity of the mutation in *β*G, and propagating to a cluster of residues in *β*B, *β*C, and a proximate segment of *β*J. Accompanying the increase in disorder in the listed segments, a slight stiffening of the protein backbone is observed in the helical region and in *β*F in both mutants. Accordingly, each of the two mutations induces subtle but notable changes in the ps-ns motion of specific regions of the backbone. Importantly, mutation-induced changes in fast fluctuations are communicated between segments in the vicinity of the two binding sites in both mutants.

### 2.4. Relaxation Dispersion Analysis

To explore the connection between the macroscopic characteristics of bile salt binding and slow internal dynamics in the protein, relaxation dispersion (*R_ex_*) NMR measurements [48,49] were performed on Q51A and Q99A hI-BABP at a static magnetic field of 14.1 T and compared to previously reported results on the WT protein [35]. As judged by the number of residues exhibiting values of *R_ex_* > 2 Hz, contribution of conformational fluctuations to transverse relaxation on the timescale amenable to CPMG *R_ex_* measurements (~0.3–10 ms) is most pronounced in WT and Q99A, whereas in Q51A they become diminished throughout the protein. In Q99A, slow motions of the protein backbone remain prevalent in the C-terminal half, but in comparison to wild-type, decrease substantially in *β*A, *β*B, and the helical region (Figure 5). Representative transverse relaxation dispersions as a function of CPMG field strength as determined for WT, Q51A, and Q99A *apo* human I-BABP at 283 K are plotted in Figure 6. Experiments have also been performed at 287 K and 291 K. Dispersion profiles were first individually fit assuming a two-state exchange process between a ground and an excited state as described in Materials and Methods. As we have reported earlier for WT [35], the dispersion profile of residues in this temperature range can be grouped into two clusters with slightly different exchange rates and excited state populations justifying two separate global fits for processes A ↔ B and A ↔ C. While residues undergoing the faster A ↔ B process are primarily located in the EF- and GH-regions (cluster I), residues involved in the slower A ↔ C process are located in *β*B, *β*D, the C/D-turn, and the helical cap (cluster II). Unlike in WT human I-BABP, motion on the ms time scale in the N-terminal half of Q99A ceases and residues with a non-flat R_ex_ profile in segments of the DEFGHIJ *β*-strands fit well to a single two-state A ↔ B exchange process with *k_ex_* = 871 ± 63 s^−1^ (283 K), 1159 ± 57 s^−1^ (287 K), 1546 ± 71 s^−1^ (291 K) and *p_b_* = 3.1 ± 0.2% (283 K), 3.5 ± 0.2% (287 K), 4.1 ± 0.2% (291 K) (Table 2). While the exchange rates and populations in the mutant are highly similar to cluster I of the wild-type protein, the chemical shift differences between the ground and the excited state (Appendix A) are considerably smaller at specific residues of the EFGH *β*-strands (e.g., T73, T78, V91, H98, and S101) suggesting that the two states are less distinct in Q99A than in WT in this region (Figure 7). Importantly, *β*D, which in the wild-type form is part of cluster II and fluctuates together with *α*-I, *β*B, and the C/D-turn, joins the EFGH-region and fluctuates at a significantly faster rate than in wild-type. Regarding the temperature dependence of the exchange process, it yields *ΔH* = 5.8 ± 0.2 kcal mol^−1^ and *ΔS* = 13.7 ± 1.2 cal mol^−1^ K^−1^ (Figure 8) showing a similar entropy-enthalpy compensation as in wild type [35]. The values of *ΔH* and *ΔS* are slightly below the ones observed in the wild-type form further indicating that the ground and excited states may be less distinct in the mutant.

### 2.5. Molecular Dynamics Simulations

MD simulations were carried out on wild-type and mutant *apo* human I-BABP variants to compare their structure, backbone dynamics, and stability. First, 1 *μ*s simulations were performed at 10 °C. Based on the *pK*_a_ values reported for WT *apo* hI-BABP [39], at pH 6.3, where the NMR measurements were performed, the most populated protonation state is when each of the three histidines (H52, H57, H98) are protonated (PPP). Accordingly, in relating MD-accessible structural and dynamic behavior to the NMR results, we first focused on this protonation state. As confirmed by the simulations, WT, Q51A, and Q99A hI-BABP are all stable and have a similar structure in the PPP state at 283 K. However, upon comparing the distance between the D and E strands along the trajectories of the three variants, which is a good measure of how open the portal for substrate entry is, significant differences in the distribution of averaged *Cα* atom-pair distances were observed (Figure 9A). The most closed conformation was exhibited by the WT protein, with more open conformations occurring at a lower fraction during the 1 *μ*s run, reflecting the dynamic nature of the molecule. Mutant Q51A showed a similar shape of distribution, with the most populated state somewhat shifted towards the more open conformations. The ratio of conformations with wider *β*D-*β*E distances (8–11 Å) was also increased. In comparison to WT and Q51A, the DE-region of Q99A proved to be more open in the highest populated conformation, however, the distribution was narrower and limited to 7.5–9.5 Å indicating a lower contribution of large fluctuations and possibly a more defined opening-closing motion of the mutant. Representative structures of the most populated states in the fully His-protonated PPP state for each variant, are depicted in Figure 9B.

To gain insight into mutation-induced changes in local flexibility along the amino acid sequence, root mean square fluctuations (*RMSF*) around the average structure reflecting the flexibility of individual residues were evaluated on the MD trajectories (283 K) in the PPP state for the two mutants and compared to that of WT (Figure 9C). The region of the N-terminal helices has proven to be less flexible in the mutants than in the WT protein and the EF-region also showed decreased flexibility. The C/D turn-region and the N-terminal half of *β*D showed increased fluctuations in Q51A. In Q99A, increased flexibility with respect to wild-type at this temperature and His-protonation state was more limited to the vicinity of the mutation. In overall, MD-derived local flexibility changes upon the mutations are in fairly good agreement with mutation-induced changes in order parameters related to the amplitude of ps-ns flexibility of N-H bond vectors reported by NMR at 283 K and pH 6.3 (Figure 4).

To investigate larger fluctuations and the stability of the variants, we subsequently carried out 1 μs MD simulations at an elevated temperature. As we have shown previously [36], although 1 μs is at the short end of the time scale of NMR-observed conformational fluctuations between the closed and the more open protein state, carrying out the simulations at elevated temperatures makes it possible to monitor local and global unfolding events of hI-BABP. Accordingly, simulations were carried out at 373 K, a temperature where the stability of the protein has previously been found to be challenged and differences between the His protonation variants in the wild-type form are well expressed in the time-frame of simulation [39]. Simulations were carried out in all possible combinations of protonated states of the three histidines (H52, H57, H98) (Figure 10).

While the WT protein mainly shows the unfolding of the EF strands and fluctuations in the helices, destabilization of the mutants in most protonated states is more extensive encompassing nearly the entire C-terminal half. This is well expressed, for example, in the double charged PEP (52p^+^57*ε*98p^+^) and PPE (52p^+^57p^+^98*ε*) states. In Q51A^PEP^, destabilization starts in the helical region, followed by the EF- and the GHIJ-region losing its structure completely around 700 ns. In Q99A^PEP^, while the helical content remains preserved near the entire length of the simulation, fluctuations in *β*D starts at an early stage (100 ns) followed by the destabilization of the DEF-region and subsequently the entire C-terminal half. When comparing with the PPP state, the additional positive charge at H57 appears to stabilize the *β*-barrel of both mutants. Similarly, protonation of H98 (when H52 and H57 are protonated also) increases the stability of not only Q99A, but also that of Q51A. The vulnerability of the PEP and PPE states is in contrast to EPP (52*ε*57p^+^98p^+^), which appears to be one of the most stable His-protonation state for WT and the mutants. Worth noting also are the differences between the uncharged, fully protonated, and single protonated states. In the EEE, EEP (52*ε*57*ε*98p^+^), and EPE (52*ε*57p^+^98ε) states mutant Q51A clearly shows a lower stability and loosened overall structure compared to WT and even more to Q99A. This is in contrast to the PEE (52p^+^57*ε*98*ε*) and PPP states, where the behavior of WT and the mutants becomes less distinct. In summary, the observed differences in secondary structure integrity and local unfolding events indicate a protonation-state dependent response to the mutations.

The introduced mutations also have an effect on the dynamics of correlated motions. This we have investigated at the level of slow motions by applying an averaging window of 5 ns on the trajectories at 283 K and clustering the together-moving regions based on the *Cα*-*Cα* distances. Previously, we have shown (350 K) that in WT hI-BABP, the EF-region moves together with the bottom of the *β*-barrel and further splitting into clusters separates the helical cap from the rest of the molecule [36]. In the present work, clustering of together-moving regions into three or four clusters (Figure 11) shows clear differences between WT and the two mutants in terms of both the separation of the EF-region alone or together with the bottom of the *β*-barrel from the rest of the molecule as well as the behavior of the helical cap. Further, as exemplified in Figure 11, the charged state of histidines affects the mutation-induced differences. Intriguingly, when four clusters were separated, the location of the fourth cluster (green color) showed clear differences between the variants. In the case of WT, it was mainly located around the D strand, while for the mutants, the N-terminal region was involved. In the case of Q99A, the extent of the fourth cluster was significantly larger than for the other variants suggesting an alteration in the separately moving regions. The observed differences in mutation- and protonated-state dependent correlated motions in hI-BABP might have a role in ligand binding and might be a subject for further MD studies.

## 3. Discussion

With the combination of positive-binding cooperativity and the site preference of di- and trihydroxy bile salts, human I-BABP is an efficient intracellular carrier recognizing structurally diverse bile salts that exist in the human body [24]. Given the nearly identical intrinsic affinities of GCA for site 1 and 2, initially it was hypothesized that site-selectivity in the heterotypic GCDA/GCA complex arises from differences in cooperativity, i.e., that GCA, the trihydroxy bile salt, can exert its high cooperative effect through site 2 only [25].

However, subsequent mutagenesis and calorimetric studies revealed no correlation between losses in site-selectivity and losses in macroscopic binding cooperativity indicating that the two phenomena are not linked as closely as thought [22]. As shown by the binding parameters of mutated hI-BABP variants, enthalpy-entropy compensation is a dominant feature of BABP-bile salt interaction. From NMR relaxation studies of BABPs and other members of the iLBP family, it is also evidenced that order-disorder transitions and conformational exchange processes have a key role in bile salt recognition. To improve our understanding of binding cooperativity and site-selectivity in BABPs, we have subjected two functionally impaired hI-BABP mutants to multiple timescale motional analysis in the *apo* state using NMR relaxation methodologies and MD simulations. Both residues are part of the binding pocket and are involved in multiple H-bond interactions stabilizing the bound bile salts in the *holo* form (Figure 1C) [19]. Among the two, mutation Q51A abrogates the site-selectivity of both GCA and GCDA while leaving binding cooperativity intact, whereas Q99A abrogates the positive binding cooperativity of GCDA without affecting site-selectivity [22]. Both mutated residues are located in the vicinity of highly dynamic turn regions, which undergo large conformational changes upon ligand binding (Figure 1B) [19]. The C/D-turn (H52-T58) proximate to Q51 moves toward the helical cap in the bound state and together with *α*-II closes the gap between the *β*-barrel and the helices. The G/H-turn (F94-N96) proximate to Q99 moves away from the EF-region accompanied by the formation of more tight contacts between *β*H and *β*I upon bile salt binding. The C/D- and G/H-turns together with the helical cap and the EF-region are thought to undergo a concerted motion in the *apo* state aiding bile salt entry into the enclosed binding pocket [19].

Our CD spectroscopic measurements show that wild-type hI-BABP and the two mutants have identical secondary structure and similar overall stability and unfolding profile, with slightly decreased melting temperature for the mutants. This is in agreement with the absence of large chemical shift changes upon the mutations and suggests that functional imperfections might have a dynamic origin. Our NMR relaxation measurements support this hypothesis and provide new indications that fluctuations on the ms timescale in hI-BABP are associated with a network of hydrophilic interactions, which acting as a scaffold brings together side chains in appropriate positions endowing the protein with remarkable plasticity and specificity at the same time. Accordingly, besides the mediation of ligand entry, internal motions likely to have implications for the site selectivity of di- and trihydroxy bile salts and binding cooperativity as well. Importantly, our results demonstrate that a single-residue mutation at either of the two mutated turn-regions has long-range effects on protein dynamics in the *apo* state. Upon the mutation at the beginning of *β*H (Q99A), the hydrophilic network is disturbed in the vicinity of site 1 near the tentative EFGH portal region. Both the enthalpy and the chemical shift difference between the ground and the low-populated excited state suggest that the structural difference between the two states becomes slightly smaller upon the mutation. A full structure elucidation of the mutant should shed more light on the differences from wild-type at the atomic level, but a plausible explanation is that changing the glutamine to an alanine disrupts an H-bonding network resulting in a “looser” EFGH-region, making the ground state closer to the hypothesized higher energy EFGH open-state. Supporting this idea, mutant Q99A shows a more open conformation in MD simulation at 283 K with increased distance between *β*D and *β*E (Figure 9AB). An increased disorder in the region is also indicated by our NMR relaxation measurements directed at faster motions revealing an increase in the ps-ns flexibility of βE (Figure 4B and Figure 12B). A more open conformational arrangement at the postulated portal in the ground state of Q99A may facilitate the entry of bile salts and can explain the faster kinetics of bile salt binding observed by stopped-flow fluorescence measurements in comparison to wild-type [33]. A looser, more plastic conformation in the vicinity of binding site 1 may also result for the first GCDA molecule to be trapped in an ‘overoptimized’ geometrical arrangement explaining the favorable enthalpic contribution of the first binding step, a marked decrease in *K_d_*_1_, and the loss of positive binding cooperativity [22]. This is in agreement with the observed chemical shift perturbation of bulky hydrophobic residues (I71, V83, L90, V92, F94, Y97) in *apo* Q99A near site 1 (Figure 3B and Figure 12B) suggesting a structural rearrangement in the region.

The observed slow dynamic changes in Q99A highlight the role of *β*D in the stabilization of site 1 in relation to the impaired positive cooperativity. As a result of the mutation, it becomes decoupled from the cluster in the N-terminal half and fluctuates together with the EF-region and the C-terminal *β*-sheet at a faster rate. In parallel, fluctuation of cluster II in the N-terminal half ceases. This suggests that interactions of *β*D likely have a specific role in driving the ms dynamics of cluster II observed in wild-type as well as in the negatively correlated motion of the helical + BCD- and EFGH-regions indicated by MD simulations [36]. Importantly, *β*D hosts residues, which have been shown to have (i) a crucial role in the communication between the two sites in hI-BABP (N61) [22] and (ii) which participate in conserved pairwise interactions stabilizing the *β*-barrel in iLBPs [36]. Among them, F63 of *β*D together with F47 of *β*C have been proposed to be part of a folding initiation site in human I-BABP [50] and belong to a cluster in I-FABP, which has been suggested to act as a nucleation center for the propagation of *β*-strands [51]. We note that the DE-region in iLBPs is unique in the sense that there are no H-bonds between the backbone atoms of adjacent strands and the source of stabilization is limited to van der Waals contacts between hydrophobic side chains. In human I-BABP, van der Waals contacts between residues F63, V65, I71, F79, and V83, are the primary source of stabilization in the region, most of which undergo a structural and/or dynamic change in Q99A suggesting a redistribution of enthalpic and entropic contributions of stabilization upon the mutation.

The strong intrinsic affinity of GCDA for site 1 might have a role in the site preference of bile salts in the mixed GCDA/GCA system in hI-BABP. In that case, structural changes are expected to be present near site 1 in mutants lacking site-selectivity. Indeed, together with the E/F-turn (Y73, M74, K77), residues F63 and V92, forming hydrophobic contacts with the steroid rings of GCDA in the bound state, show above average chemical shift changes in *apo* Q51A (Figure 3A and Figure 12A), suggesting a structural rearrangement in the region, far away from the site of the mutation. Molecular dynamics simulations performed on various homo- and heterotypic complexes of hI-BABP with di- and trihydroxy bile salts also indicate that by allowing a better closure of the helical cap, a deep penetration of the more hydrophobic dihydroxy bile salt into the protein core at site 1 may have a key role in site-selectivity [52]. Of note, several of the affected hydrophobic residues near site 1 are part of a hydrophobic patch that has been found to be exposed in a partially unfolded hI-BABP state [36] and overlap with residues thought to be implied in ligand-mediated nuclear export signalling [53] and the stimulation of nuclear receptors [4,54].

Similar to mutation Q99A, besides the structural perturbations at distant sites indicated by chemical shift changes, disruption of the hydrophilic network in the vicinity of the C/D-turn by mutation Q51A has long-range dynamic effects. Specifically, the observed changes in fast dynamics show that perturbations caused by the mutation are propagated to the EF- and GH-regions, *β*J, and the helical cap. Moreover, ms motions observed in wild-type in both the C- (cluster I) and the N-terminal half (cluster II) cease nearly entirely upon the mutation. We note that while in WT and Q99A, the position of residues exhibiting conformational exchange according to the model-free analysis (*μ*s-ms timescale) generally fell into the regions exhibiting a nonflat dispersion profile in CPMG relaxation dispersion experiments (0.3–10 ms timescale), in Q51A, Model-free reported additional residues in the helical cap, *β*D, and *β*J. This indicates that while in WT and Q99A the CPMG *R_ex_*-reported global motion is the dominating exchange process, in Q51A *μ*s fluctuations (outside of the CPMG *R_ex_*-sensitive timescale) are likely to exist. Channeling of ms fluctuations into faster motions in Q51A decreases the stabilizing interactions of the E/F-turn region with *β*D and *α*-I, increasing the disorder in the upper segment of the binding cavity (Figure 12A). As noted above, dynamic changes are accompanied by moderate chemical shift changes in the region indicating that redistribution of motional freedom is related to a structural reorganization. This suggests that concerted slow motions involving residues in the upper segment of the binding cavity may have a role in the positioning of specific amino acid side chains required for the site-selectivity of bile salts with different hydroxylation pattern in the heterotypic complex. Specifically, according to the high-resolution structure of the hI-BABP:GCDA:GCA complex [19], the side-chain of Q51 forms a hydrogen-bond with the side-chain of N61 (*β*D) at site 1, which in turn interacts with T73 of the E/F-turn (Figure 1C). The position of T73 is stabilized by another H-bond with K77 and the interaction with the bound GCDA at site 1. The E/F-turn is additionally involved in long-range interactions with *α*-I enabling a dynamic coupling between the pliable EF-strands and the helical cap closing the *β*-barrel. Q51 additionally forms an H-bond with the *ε*NH of W49 located in the middle of the binding pocket as well as with the backbone atoms of M59. Nearly all of the residues involved in the interactions listed above show increased flexibility on the ps-ns timescale and a concomitant abrogation of ms motion in the *apo* form of the mutant. The role of dynamics in the site-selectivity of bile salts in hI-BABP is consistent with findings in the chicken liver analogue, where the presence or absence of a disulphide bridge linking *β*F and *β*G accompanied by a change in the rigidity of the protein backbone has been found to be able to modulate the site-preference of ligands [55].

In comparison with the chicken liver analogue [17,37], dynamics of the N-terminal *β*-sheet, primarily the BCD *β*-strands have an additional role in conformational exchange in the human ileal form [35]. Previously, we have related this observation to the presence of two additional histidines (H52, H57) in the C/D-turn besides H98 in *β*H and suggesting that the tautomerization equilibrium of histidines in the vicinity of two instead of a single turn region allows a more sophisticated regulatory mechanism of ligand recognition in the human form [39]. Our MD simulation analysis of Q51A and Q99A supports this hypothesis. While no major changes in the tautomerization equilibria of histidines are indicated upon the mutations (Appendix A), the marked effect of histidine protonation on protein flexibility shown by the simulations suggests that even subtle modulation of the stability of a charged histidine in the two turn-regions may affect conformational fluctuations involving the portal region and the binding pocket.

Intriguingly, in both mutants, but especially in Q99A, the helical region shows a slight stiffening on the ps-ns timescale (Figure 4, Figure 9C and Figure 12A,B). This may be an indication that similar to the conformational rearrangement observed upon ligand binding [19], opening of the EFGH-region in *apo* Q99A is accompanied by a closure and stabilization of the helical cap. Reduced motional freedom of the helical cap is further manifested in the cessation of slow motions in the linker connecting *α*-II to the N-terminal *β*-sheet. This is in contrast to wild-type, where the K35-I36 segment displays high ms flexibility in the *apo* state [35]. We note that a similar cessation of ms motions is observed in the linker between *α*-II and *β*B in WT at both elevated (8.0) and lower (5.4) pH [39], underlying the modulatory role of H-bonds and electrostatic interactions involving the proximate histidines in slow dynamics.

In conclusion, investigation of the two functionally impairing mutations of hI-BABP in the *apo* state shows an intimate relation of structural and dynamic effects not only in the mediation of ligand entry, but also with implications for binding cooperativity and the site-preference of bile salts with different hydroxylation patterns. The dynamic coupling between the EFGH- and the C/D-regions provides new indications that the two regions act in an anticorrelated manner alternating between a closed^EFGH^/open^CD+*α*^ (*apo*-like) and an open^EFGH^/closed^CD+*α*^ (*holo*-like) conformations mediating ligand binding. Both mutations, while differing functionally, identify the DEF region as a pliable region encoding the ‘engine’ for a change in hI-BABP function, with possessing a high degree of vulnerability as well. The identified dynamic interaction networks may become new targets in the design of pharmaceutical agents for the modulation of hI-BABP function with implications for targeting bile salt transport and bile salt-induced stimulation of nuclear receptors. Comparative analysis of internal dynamics in wild-type and mutated hI-BABPs in homo- and heterotypic complexes with various combinations of di- and trihydroxy bile salts is on its way and should shed further light on ligand recognition.

## 4. Materials and Methods

### 4.1. Sample Preparation

Site-directed mutagenesis was carried out using overlap extension polymerase chain reaction. The methods used for the expression and purification of ^15^N and ^2^H/^15^N-labeled wild-type and mutated (Q51A, Q99A) human I-BABP used in the experiments are detailed elsewhere [34]. Protein was dialyzed into a buffer containing 20 mM potassium phosphate, 50 mM KCl, and 0.05% NaN_3_ (Sigma-Aldrich, St. Louis, MI, USA) at pH = 6.3. Protein concentration was determined by absorbance at 280 nm using an extinction coefficient of 12,930 M^−1^ cm^−1^ (WT), 12,063 M^−1^cm^−1^ (Q51A), 12,445 M^−1^ cm^−1^ (Q99A) obtained by composition analysis [56].

### 4.2. CD Spectroscopy and Thermal Denaturation Experiments

CD spectroscopy experiments of WT and mutant human I-BABPs were carried out on a Jasco J-810 instrument (Japan Spectroscopic Co., Tokyo, Japan) equipped with a Peltier-controlled thermostat. CD spectra were recorded at 0.1 mg/mL protein concentration in 20 mM K-phosphate, 50 mM KCl buffer, at pH values 5.8, 6.3, and 8.0 in a quartz cell of 1 mm pathlength. 5–10 scans were accumulated with a scanning rate of 50 nm/min, 2 s data integration time and 2 nm bandwidth. Thermal denaturation was either followed at 200 nm in the temperature range of 20–80 °C with a scanning rate of 2 °C/min or complete spectra were recorded at 5 °C steps. A two-state transition model described by Shih et al. [57] showed a good fit to the fully reversible denaturation curves. Secondary structure composition was analyzed by the BeStSel webserver [41] (https://bestsel.elte.hu accessed between 1 February 2022 and 31 May 2022).

### 4.3. NMR Data Collection

Multidimensional NMR experiments were carried out on 600 MHz Varian NMR SYSTEM™ spectrometer equipped with a 5-mm indirect detection triple ^1^H^13^C^15^N resonance z-axis gradient probe. The backbone resonance assignment of *apo* human I-BABP has been determined earlier [34]. To obtain the amide ^1^H and ^15^N assignment of the mutants, the wild-type assignment has been transferred to the corresponding spectra and confirmed by 3D ^15^N-HSQC-TOCSY [58] experiments performed on uniformly [^15^N]-enriched Q51A and Q99A human I-BABP. Spectral processing was carried out using Felix 2004 (Accelrys, Inc.) and CCPNMR. The ^15^N T_1_, T_2_, and steady-state {^1^H}-^15^N NOE measurements [59,60,61] were collected on U-[^15^N]-enriched wild type and mutated human I-BABP at 283 K. Backbone amide ^15^N T_1_ values were measured from two series of eight spectra (24 transients, interscan delay of 1.5 s) with the following relaxation delay times *T* = 20, 100, 190, 290, 390, 530, 670, and 830 ms, and *T* = 20, 50, 100, 170, 240, 340, 480, and 630 ms. Amide ^15^N T_2_ values were obtained similarly with *T* = 10, 30, 50, 70, 110, 150, and 190 ms, and *T* = 10, 30, 50, 90, 130, 150, and 170 ms. Steady-state {^1^H}-^15^N NOE values were obtained in triplicate (32 transients each) by recording spectra with and without (blank) the use of ^1^H saturation applied during the last 5 s of a 7 s delay between successive transients. Presaturation was achieved with the use of 120° ^1^H pulses applied every 5 ms [62]. The RF field strength of the ^1^H hard pulse was 11.6 kHz. Relaxation dispersion data were obtained on [80% ^2^H, 99% ^15^N]-labeled protein at 283, 287, and 291 K, using a relaxation compensated Carr-Purcell-Meiboom-Gill (CPMG) dispersion experiment performed in a constant time manner [48,49]. The constant time delay (*T_CP_*) was set to 40 ms. Spectra were collected as a series of 19 two-dimensional data sets with CPMG field strengths (*υ_CPMG_*) of 25, 50, 74, 99, 123, 147, 172, 195, 219, 242, 289, 335, 380, 425, 469, 556, 641, 764, and 883 Hz. A reference spectrum was obtained by omitting the CPMG period in the pulse sequence [63]. Spectra (3 s interscan delay, 24 transients) were acquired in duplicate. Protein concentration was 0.5 mM in all experiments.

### 4.4. NMR Relaxation Analysis

Spectral densities were calculated from the ^15^N T_1_, T_2_, and steady-state {^1^H}-^15^N NOE relaxation parameters according to Abragam [64]. At the investigated temperature of 283 K, ^15^N relaxation parameters could reliably be determined for 117, 108, and 110 backbone amide positions of the 126 nonproline residues for WT, Q51A, and Q99A, respectively. Resonances showing severe overlap and those of low intensity were excluded from the analysis. Amide N-H bond lengths were assumed to be 1.02 Å and the ^15^N chemical shift anisotropy was estimated as -172 ppm during the calculation [65]. To characterize the spatial restriction of the ^15^N-^1^H bond vector on the picosecond to nanosecond time scale, the NMR relaxation data were analyzed within the extended model-free formalism [44,45,46,47]. Motional parameters have been determined using FAST-Modelfree (Facile Analysis and Statistical Testing for Modelfree [66]), interfaced with Modelfree 4.2 [67,68]. An initial estimate of the rotational diffusion tensor was calculated from the filtered T_1_/T_2_ ratios using the programs r2r1_diffusion (http://www.palmer.hs.columbia.edu/software/r2r1_diffusion.html accessed between 1 January 2021 and 30 June 2021) and pdbinertia (http://biochemistry.hs.columbia.edu/labs/palmer/software/pdbinertia.html accessed between 1 January 2021 and 30 June 2021). The criteria for inclusion of residues in the diffusion tensor estimate relied on the method by Kay et al. [69]. Coordinates for wild-type and mutated human I-BABP were obtained from PDB file 1O1U [15].

Contributions to transverse relaxation rates of conformational exchange in CPMG relaxation dispersion measurements were analyzed assuming a two-state exchange process using the all-timescales multiple quantum Carver-Richards-Jones formulation [70] implemented in GUARDD [71].

### 4.5. Molecular Dynamics Simulations

Molecular dynamics simulations were carried out as it was described earlier [36,39] using the GROMACS package [72] on the solution structure of human I-BABP (PDB: 1O1U [15]). Mutations Q51A or Q99A were built in the structure by using Swiss-PdbViewer [73]. AMBER-ff99SB*-ILDNP force field [74] and TIP4P parametrization [75] were used in the simulations. The total charge of the system was neutralized, and the physiological salt concentration was set by placing Na^+^ and Cl^−^ ions. Energy minimization of starting structures was followed by relaxation of constraints on protein atoms in three steps and an additional NVT step (all of 200 ps). 1 μs trajectories of NPT simulations at 283 K and 370 K at 1 bar were recorded. Secondary structure compositions of the frames of MD trajectories were determined by the DSSP algorithm [76]. A series of simulations were carried out by varying the protonation levels of the three histidine residues (H52, H57, and H98), setting them individually as protonated at N*ε*2, or N*δ*1, or at both nitrogens (positively charged form). Molecular graphics was performed with the UCSF CHIMERA package (University of California, San Francisco, CA, USA) [77]. The distance between the D and E strands was defined as the average distance of four backbone *Cα* atom pairs along the two strands (residues M59-T73, T60-Q72, N61-I71, and K62-N70). The first 100 ns of the trajectories were excluded from the analysis. Root mean square fluctuation (RMSF) around the average MD structure was calculated by the GROMACS package. For the clustering of correlated motions in the molecules, we used an in-house program to find out which regions of the molecule move together as it was described earlier [36]. The calculation was based on the clustering of the time series of alterations of pairwise distances between *Cα* atom positions. To focus on the slow movements of the system, fast motions were averaged by 5 ns (i.e., 250 frames) to determine the *Cα* atom positions before the calculation. Ignoring the first 100 ns, correlations were calculated for each *Cα*—*Cα* distance and then these values were clustered using Matlab kmeans clustering function and visualized using UCSF CHIMERA [77].

## Figures and Tables

**Figure 1 ijms-23-11346-f001:**
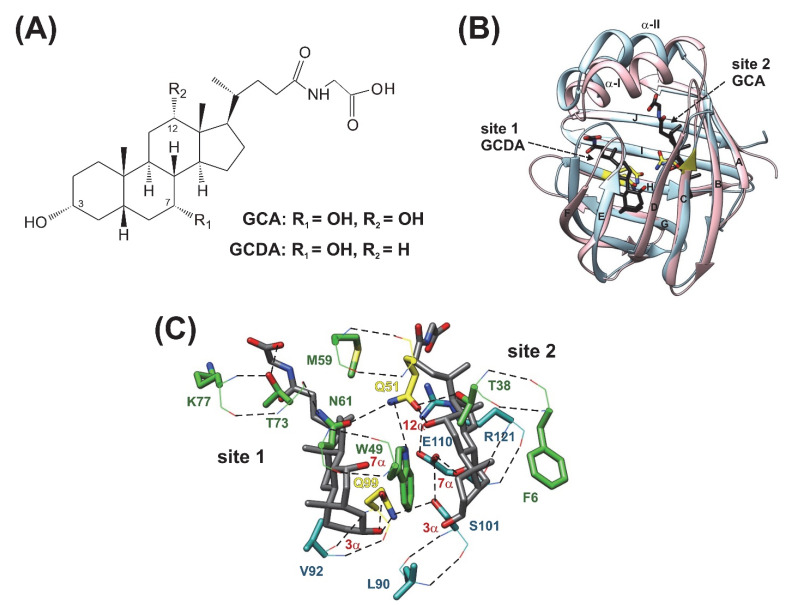
(**A**) Chemical structure of glycocholic acid (GCA) and glycochenodeoxycholic acid (GCDA), the two most abundant bile salts in humans. (**B**) Superimposed ribbon diagrams of *apo* human I-BABP (PDB: 1O1U, blue) [15] and the heterotypic doubly-ligated human I-BABP:GCDA:GCA complex (PDB: 2MM3, pink) [19]. Mutated residues investigated in this work are indicated in yellow. Bound bile salts are shown in black. The most representative member of the lowest energy structural ensemble is shown for both the *apo* and the *holo* forms. (**C**) Stabilizing H-bond interactions in the binding pocket of the heterotypic doubly-ligated hI-BABP complex. Residues involving H-bond networks including Q51 and Q99 are shown in green and cyan, respectively. H-bonds are depicted in black. The two mutated residues are indicated in yellow. Backbone and side chain atoms are depicted in lines and stick, respectively.

**Figure 2 ijms-23-11346-f002:**
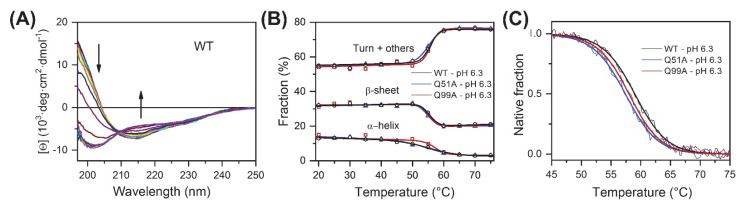
CD spectroscopy of *apo* human I-BABP variants. (**A**) CD spectra of the WT protein recorded as a function of temperature from 10 °C to 85 °C at 5 °C steps. Arrows show the direction of spectral changes. Temperature dependence for all variants is shown in Appendix A. (**B**) Secondary structure composition was analyzed using the BeStSel method [40,41] as a function of temperature for wild-type hI-BABP and the two mutants. Secondary structure is summed up in three groups, *α*-helix, *β*-sheet, and turn+others. (**C**) Overall stability of the variants at pH 6.3 was investigated and compared by thermal denaturation using a continuous heating at a rate of 2 °C/min and followed by the CD signal at 200 nm, which reflects the changes in the *β*-sheet and disordered structure of the molecule. Thermal stability experiments for pH 5.8, 6.3, and 8.0, are presented in Appendix A.

**Figure 3 ijms-23-11346-f003:**
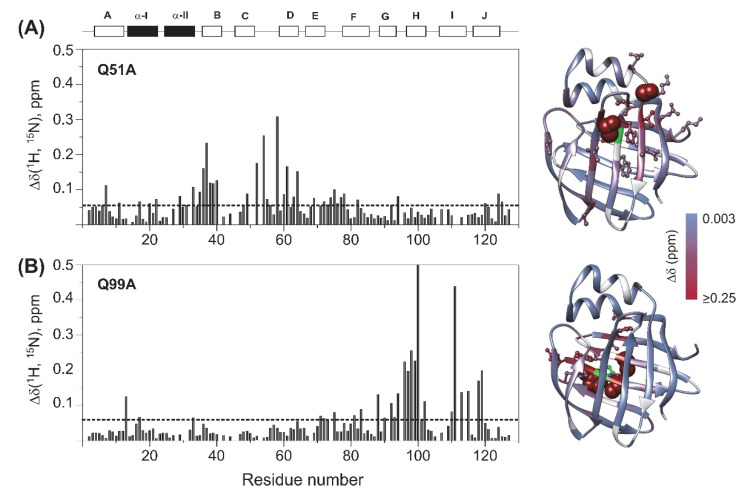
Backbone amide chemical shift differences between wild-type and (**A**) Q51A, (**B**) Q99A *apo* human I-BABP (283 K). Combined (^1^HN, ^15^N) chemical shifts were calculated using the equation of *Δδ*_1*HN,* 15*N*_
*=√[(Δδ_HN_)*^2^*+(w*_1_**Δδ_N_)*^2^*]*, where *w*_1_ (=0.154) is a weight factor determined using the BioMagResBank chemical shift database [42]. The dashed line corresponds to the mean value. Secondary structural elements are indicated at the top. Values of *Δδ*_1*HN,*15*N*_ are mapped on the ribbon diagram of wild-type human I-BABP (PDB: 1O1U [15]) in a blue-to-red gradient. Residues exhibiting *Δδ*_1*HN,*15*N*_ > 0.25 ppm (arbitrary limit) are highlighted as spheres, whereas those exceeding the mean by more than half of the standard deviation are shown in a ball-and-stick representation. The position of the mutations is indicated in green. Residues with no available data are colored in gray.

**Figure 4 ijms-23-11346-f004:**
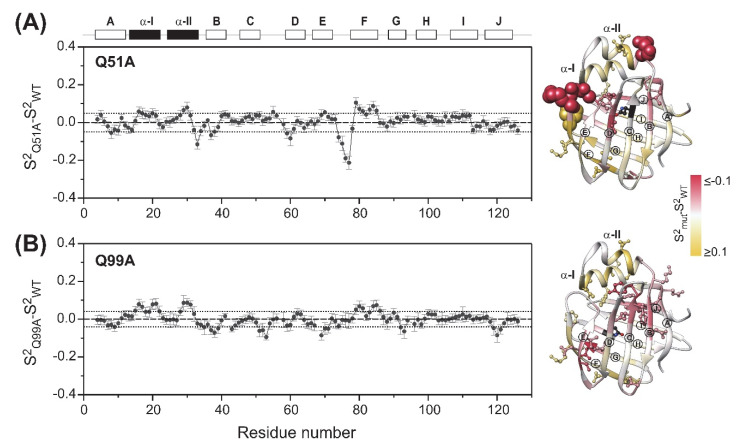
Differences in the ps-ns flexibility of the protein backbone between wild-type and (**A**) Q51A, (**B**) Q99A *apo* human I-BABP (283 K) as determined from ^15^N relaxation measurements. Differences in the generalized order parameter (*S*^2^*_mut_-S*^2^*_WT_*) obtained from MF analysis were averaged for a window of three consecutive residues (*ΔS*^2^***) and shown as a function of the amino acid sequence. Secondary structural elements are indicated at the top. Values of *ΔS*^2^*** mapped on the ribbon diagram of wild-type human I-BABP (PDB: 1O1U [15]) in a yellow-to-gray-to-red gradient are shown on the right. Residues exhibiting |*ΔS*^2^*| > 0.1 ppm (arbitrary limit) are highlighted as spheres, whereas those exceeding the mean by more than half of the standard deviation (dotted line) are shown in a ball-and-stick representation. *β*-strands are labeled for better readability. The position of the mutations is indicated in black.

**Figure 5 ijms-23-11346-f005:**
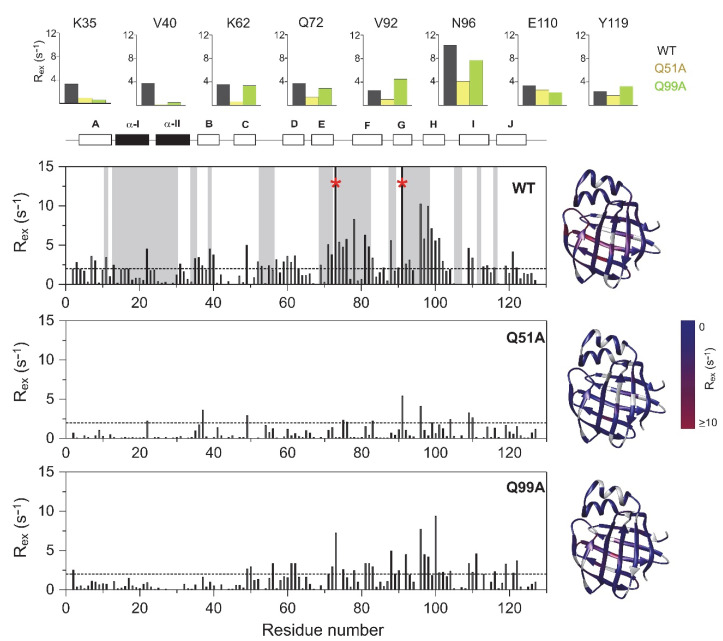
Values of ^15^N *R_ex_* (at a static magnetic field strength of 14.1 T) of ^2^H/^15^N-labeled wild-type (**top**), Q51A (**middle**), and Q99A (**bottom**) *apo* human I-BABP at 283 K (pH = 6.3) as a function of amino acid sequence. *R_ex_* was estimated from the difference in *R*_2*eff*_ at the lowest and highest ν_CPMG_ values. Residues displaying a *C_α_* positional difference between the *apo* and the heterotypic double-ligated *holo* form larger than 2 Å are shown in a gray background. Residues exhibiting a *R_ex_* > 15 Hz are marked with a red asterisk. Secondary structural elements are indicated at the top. For representative residues, ^15^N *R_ex_* in the investigated human I-BABP variants are shown in large at the top (WT: black, Q51A: yellow, Q99A: green). Values of *R_ex_* are mapped on the ribbon diagram of wild-type human I-BABP (PDB entry 1O1U [15]) in a blue-to-red gradient on the right for each variant.

**Figure 6 ijms-23-11346-f006:**
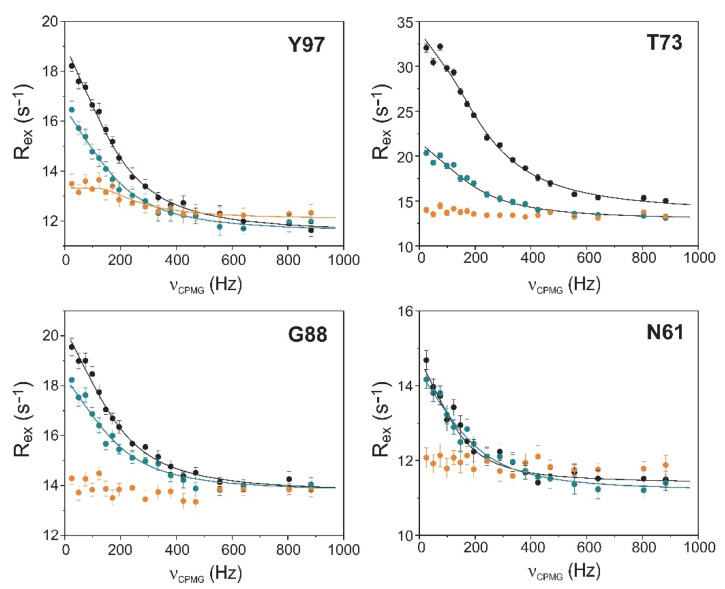
Transverse relaxation dispersions of the backbone ^15^N nuclei of selected amino acid positions in wild-type (black), Q51A (orange), and Q99A (green) *apo* human I-BABP as a function of CPMG *B*_1_ field strength. The experiments were performed at 283 K at a static magnetic field of 14.1 T. Error bars are standard deviations derived from duplicate measurements. Solid lines correspond to global two-state exchange models with parameters listed in Table 2.

**Figure 7 ijms-23-11346-f007:**
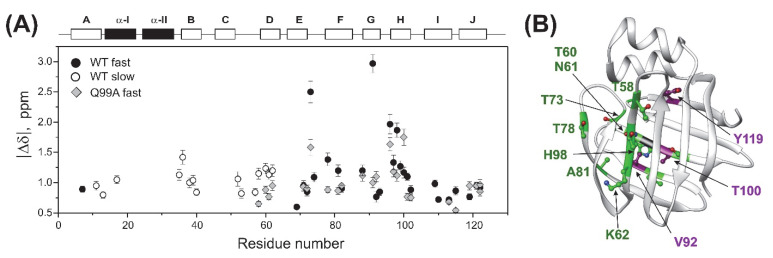
Backbone ^15^N chemical shift differences between the ground and higher energy states. (**A**) Values of |*Δδ*| along the amino acid sequence as derived from CPMG relaxation dispersion measurements on wild-type (circle) and Q99A (quadrangle) *apo* human I-BABP with parameters listed in Table 2. For residues involved in global fluctuation at faster (cluster I) and slower (cluster II) exchange rates, values of |*Δδ*| are depicted in closed and empty symbols, respectively. Secondary structure elements are shown at the top. (**B**) Residues exhibiting a decrease (green) or increase (purple) in |*Δδ*| exceeding 20% in the mutant relative to the wild-type form are highlighted on the ribbon diagram of *apo* human I-BABP (PDB: 1O1U [15]). The position of the mutation is indicated in black.

**Figure 8 ijms-23-11346-f008:**
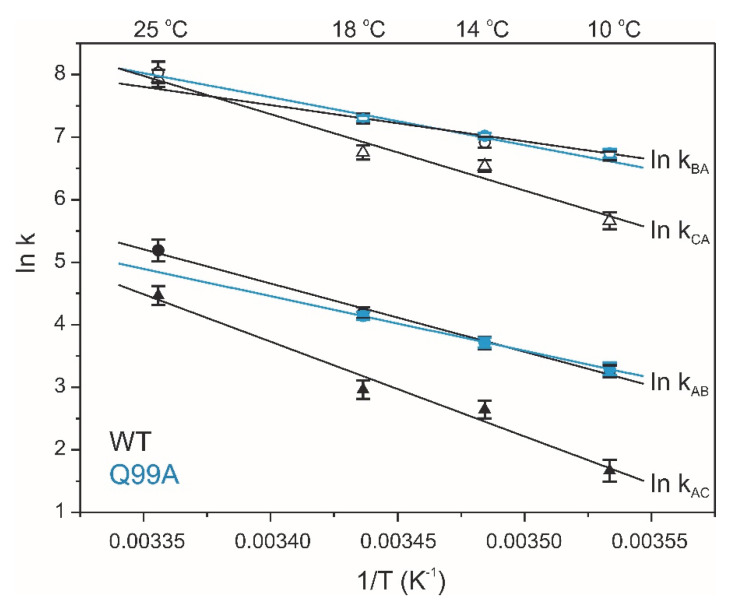
Temperature dependence of conformational exchange in wild-type (black) and Q99A (cyan) *apo* human I-BABP. Forward (closed) and reverse (open) rate constants of the conformational exchange obtained from global two-state analysis of relaxation dispersion curves. The slower cluster (triangles) is observed only in the wild-type form.

**Figure 9 ijms-23-11346-f009:**
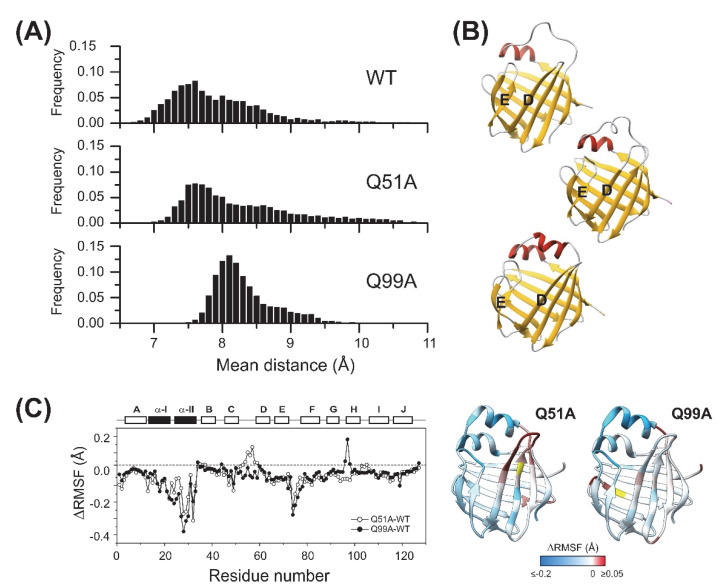
Structural and dynamic differences between the investigated *apo* human I-BABP variants as revealed by MD simulations at 283 K. The openness of the ligand entry portal of *apo* hI-BABP was inspected by the distance between the D and E *β*-strands. (**A**) Distributions of the average distance between backbone *Cα* atom-pairs of the two *β*-strands are shown (see *Methods*). (**B**) Representative structures for the most populated conformations regarding the D-E distance for the three variants. Q99A exhibits a more open conformation than WT and Q51A along the MD trajectories. (**C**) Differences in *RMSF* values between the mutants and WT *apo* hI-BABP at 283 K. Results are shown for the most populated protonation state at pH 6.3 with all three histidine residues are protonated. Values of *ΔRMSF* are mapped on the ribbon diagram of *apo* hI-BABP (PDB: 1O1U [15]) for the two mutants in a blue-to-white-to-red gradient with blue and red colors corresponding to decreased and increased flexibility, respectively, relative to wild-type. The position of the mutations is indicated in yellow.

**Figure 10 ijms-23-11346-f010:**
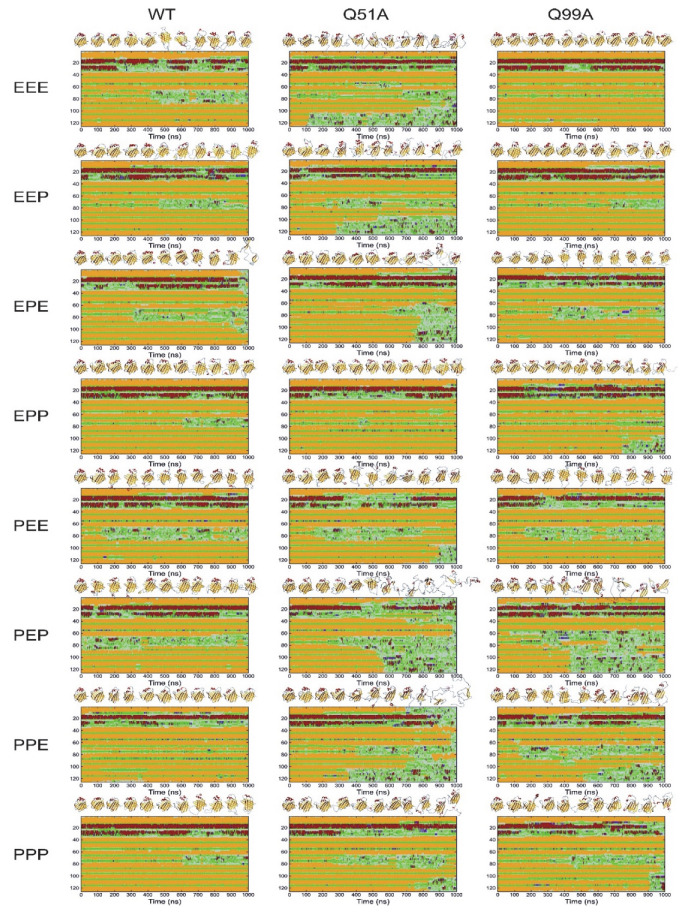
Comparison of the stability and flexibility of WT, Q51A, and Q99A *apo* hI-BABPs by 1 *μ*s MD simulations at 373 K in all possible combination of histidine protonated states (E: uncharged, P: cationic, e.g., PPP indicates that His residues H52, H57, and H98, are all protonated). Secondary structure composition is presented along the sequence as a function of time (color coding: red: *α*-helix; purple: 3_10_-helix, yellow: *β*-strand, green: turn, gray: irregular). 3D structures corresponding to every 100 ns are presented at the top of the panels.

**Figure 11 ijms-23-11346-f011:**
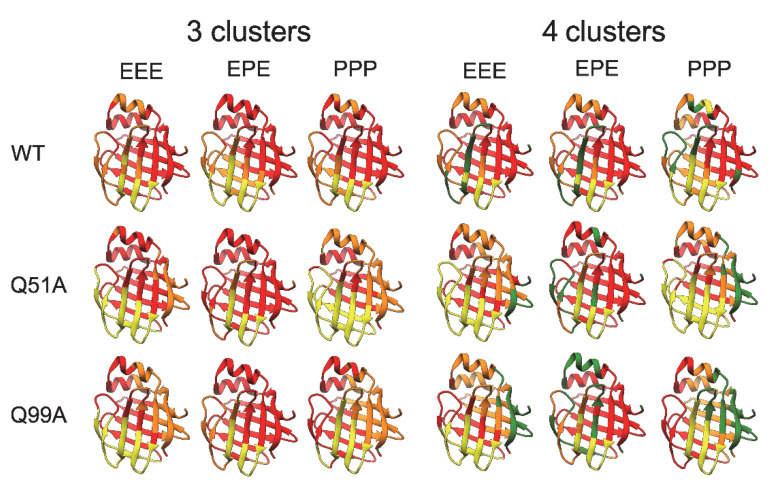
Correlated motions in the investigated *apo* human I-BABP variants as revealed by MD simulations for different protonation states (E: neutral, P: cationic His side chain of residues H52, H57, and H98). Clustering of regions for ‘slow’ correlated motions at 283 K was based on the analysis of alterations of pairwise distances between *Cα* atoms and splitting the residues moving together into three or four clusters. Clusters are represented by different colors on the ribbon diagram of *apo* human I-BABP (PDB: 1O1U [15]).

**Figure 12 ijms-23-11346-f012:**
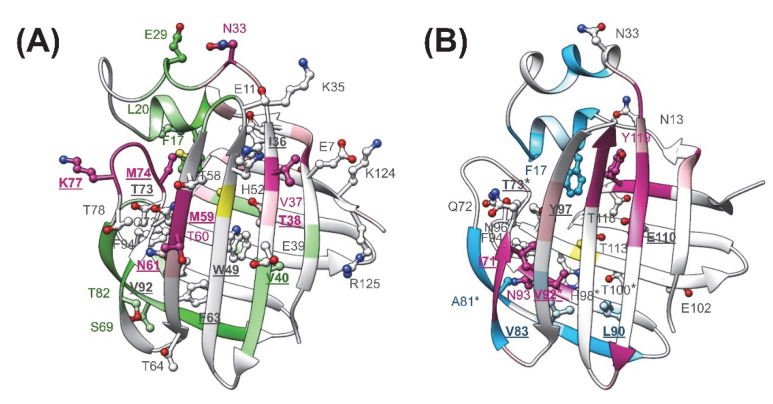
Structural and dynamic perturbations as a result of the mutation in the *apo* form of (**A**) Q51A and (**B**) Q99A are mapped onto the ribbon diagram of the corresponding MD-derived structure. Residues with increased flexibility on the ps-ns time scale in comparison to wild-type are shown in magenta for both mutants. Residues with decreased ps-ns flexibility are depicted in green (Q51A) or cyan (Q99A). Residues exhibiting above average chemical shift change as a result of the mutation are depicted in a ball-and-stick representation. Position of the mutations is indicated in yellow. Residues involved in H-bond interactions or in hydrophobic interactions with the bound bile salts in the *holo* form are underlined and shown in bold. Residues in Q99A human I-BABP with altered (>25%) chemical shift difference between the ground and the sparsely populated higher energy state inferred from the CPMG measurements are marked with an asterisk.

**Table 1 ijms-23-11346-t001:** Summary of ^15^N NMR spin-relaxation rates, generalized order parameters, and the number of residues with *τ_e_* and *R_ex_* contributions according to Model Free analysis of ^15^N backbobe relaxation measurements for the investigated human I-BABP variants (283 K). Physical quantities are italicized.

	WT	Q51A	Q99A
^15^N *R*_1_ (s^−1^)	1.4 ± 0.2	1.5 ± 0.2	1.4 ± 0.1
^15^N *R*_2_ (s^−1^)	14.0 ± 1.3	13.3 ± 1.2	13.7 ± 1.8
{^1^H} ^15^N *NOE*	0.80 ± 0.06	0.80 ± 0.07	0.79 ± 0.06
*τ**_m_* (ns)	9.6	9.3	9.4
*D* _||_ */D* _⊥_	1.2	1.1	1.1
*S* ^2^	0.92 ± 0.07	0.91 ± 0.07	0.92 ± 0.06
Number of residues			
model 1 (*S*^2^)	78	72	78
model 2 (*τ_e_*)	12	18	14
model 3 (*R_ex_*)	30	9	14
model 4 (*R_ex_*, *τ_e_*)	-	2	-
model 5 (*S_f_*^2^, *S_s_*^2^)	2	2	3

**Table 2 ijms-23-11346-t002:** Kinetic and thermodynamic parameters of conformational exchange in the investigated human I-BABP mutants deduced from ^15^N backbone relaxation dispersion NMR measurements. Parameters obtained previously for WT are given in the footnote. Physical quantities are italicized.

(A)		Q99A	
global fit analysis	283 K ^a^	287 K ^b^	291 K ^c^
*k_ex_* (s^−1^)	871 ± 63	1159 ± 57	1546 ± 71
*p_B_* (%)	3.1 ± 0.2	3.5 ± 0.2	4.1 ± 0.2
*k_AB_* (s^−1^)	27 ± 3	41 ± 3	63 ± 4
*k_BA_* (s^−1^)	844 ± 98	1118 ± 106	1483 ± 123
*Δ**G_AB_* (kcal/mol)	2.0 ± 0.1	2.0 ± 0.1	1.9 ± 0.1
(B)		Q51A	
individual fit analysis (283 K)	*k_ex_* (s^−1^)	*p_E_* (%)	*R_ex_* (Hz)
Gly22	364 ± 48	0.8 ± 0.3	2.8 ± 0.2
Val37	25 ± 6	6 ± 2	1.7 ± 0.4
Trp49	768 ± 309	0.3 ± 0.4	2.5 ± 0.6
Gly75	35 ± 21	5 ± 1	1.8 ± 0.3
Gly76	112 ± 30	2 ± 0.7	2.4 ± 0.3
Val91	979 ± 57	0.9 ± 0.2	5.9 ± 0.2
Tyr97	27 ± 15	3.3 ± 0.9	0.9 ± 0.1
Val104	1501 ± 304	0.2 ± 0.3	2.1 ± 0.5
Val109	2148 ± 963	0.4 ± 0.9	2.3 ± 0.9

^a^ WT: cluster I *k_ex_* = 836 ± 59 s^−1^, *p_b_* = 3.1 ± 0.2%, cluster II *k_ex_* = 294 ± 40 s^−1^, *p_b_* = 1.8 ± 0.2%; ^b^ WT: cluster I *k_ex_* = 1049 ± 88 s^−1^, *p_b_* = 3.9 ± 0.2%, cluster II *k_ex_* = 705 ± 69 s^−1^, *p_b_* = 2.0 ± 0.2%; ^c^ WT: cluster I *k_ex_* = 1540 ± 120 s^−1^, *p_b_* = 4.3 ± 0.2%, cluster II *k_ex_* = 883 ± 103 s^−1^, *p_b_* = 2.2 ± 0.1%.

## Data Availability

Not applicable.

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
