# Peer review of "Multiple Timescale Dynamic Analysis of Functionally-Impairing Mutations in Human Ileal Bile Acid-Binding Protein"

_ijms, 2022, doi:10.3390/ijms231911346_

Round 1
Reviewer 1 Report
The manuscript offers new insights on the molecular basis of the observed loss of function of two mutants of human ileal bile acid-binding protein, through a residue-level accurate analysis of the changes in protein dynamics, based on NMR and MD approaches. The observed changes in mutant protein dynamics are essential to understand the basis of binding site-selectivity, cooperativity and highlight the coupling between two important regions (EFGH- and the C/D-region). These regions, acting in an anticorrelated manner and alternating between a closedEFGH/openCD+α (apo-like) and an openEFGH/closedCD+α (holo-like) conformations, mediate ligand binding. The reported results are very interesting, however the manuscript should be improved addressing the following issues.
1) Page 5, line 154: “the α-helices, which correspond to the N-terminal “cap” of hI-BABP, loosen up at lower temperatures, well before the global unfolding transition”. This is an interesting point, the midpoint of alpha-helices unfolding transition observed for WT and Q51A could be quantified by fitting the curves reported in Figure 2B.
2) Page 5, lines 162-164: to increase the manuscript readability, the effect of the different pHs on the shift of population from a closed to a more open protein state should be better described, here or in the introduction section. Which pH favours the open state?
3) Page 7, line211: The analysis of the changes in backbone motions has been performed for the whole proteins, what the authors mean with “the two turn regions”?
4) Section 2.3 lines 214-217 and 235-242. Some of the reported experimental details should be moved to Materials and Methods section
5) Page 8, lines 248-253: The comparison with CPMG results should be moved in the Discussion section, after presenting relaxation dispersion analysis results
6) Page 9, Figure 4. Please specify the criterion to fix a threshold to 0.1. The threshold employed to select ball and stick residues as significant should be represented as a line on the graph. Please label the strands on the 3D structure to increase figure readability.
7) Page 10, line 321. This reviewer believes that the term “slow cluster” is not correct, as it is the exchange process that is slower. Legend of Figure 7 could be rephrased as “residues involved in fast and slow exchange processes”.
8) Page 16, lines 440-451. The identification of correlated motions is an important issue, however this section of the manuscript is not clear enough. The last paragraph of 2.5 section needs a revision of English language. The clustering method employed to highlight correlated motions should be better described in the results and in the Material and Methods section. A deeper description of Figure 11 is needed to clarify which are the basis of the conclusions drawn from this analysis.
9) Page 18-21 Revision of English language is required in some sections. In particular, in the Discussion section, the meaning of some sentences is not clear, some examples:
line 553 “The strong intrinsic affinity of GCDA for site 1 raises the possibility that it (?) may have a role in the site preference of bile salts in the mixed GCDA/GCA system in hI-BABP”. Please clarify this point.
lines 568-569 “Similar to Q99A, besides the observed chemical shift changes at distant sites, disruption of the hydrophilic network in the vicinity of the C/D-turn by mutation Q51A has long-range dynamic effects as well.”
Minor remarks:
1) Figure 1 legend, line 45: “Residues involved in H-bond networks involving...…” instead of “H-bond networks
Involving…”. Specify that H-bonds are depicted in black
2) Page 7, line 205 and 206: “in the linker” instead of “a linker”
3) Page 8, line 265 “observed” instead of “indicated”
4) Temperature units should be the same (Kelvin or °C) through the manuscript
5) Page 10, Figure legend 5: “Residues exhibiting a Rex > 15 Hz are marked with a cross”. The cross is not visible
6) Supplementary Material, Page 1: Bence Balterer name is missing
Author Response
We thank the reviewer for his or her valuable comments and suggestions. Below we list our point-by-point responses:
1) Page 5, line 154: “the α-helices, which correspond to the N-terminal “cap” of hI-BABP, loosen up at lower temperatures, well before the global unfolding transition”. This is an interesting point, the midpoint of alpha-helices unfolding transition observed for WT and Q51A could be quantified by fitting the curves reported in Figure 2B.
Response
We thank the reviewer for his or her comment. We have calculated the midpoint temperatures for the unfolding of helices of the I-BABP variants and added to the results (pg. 5, lines 168 and 179). For WT, Q51A, and Q99A we obtained 53.8 ± 0.7 °C, 53.0 ± 0.8 °C, and 56.3 ± 0.8 °C, respectively (fitted by a two-state unfolding model).
2) Page 5, lines 162-164: to increase the manuscript readability, the effect of the different pHs on the shift of population from a closed to a more open protein state should be better described, here or in the introduction section. Which pH favours the open state?
Response
We thank the reviewer for bringing this into our attention. We agree that the effect of pH on the equilibrium between the closed and the more open state should be discussed in more detail at the beginning of the manuscript. Therefore, a few sentences have been inserted into the corresponding paragraph in the Introduction (pg. 4):
“Cessation of slow motion upon either bile salt binding or lowering the pH in the apo form, as revealed by Rex measurements, indicates similarities in the dynamic behaviour of the complexed and the histidine-protonated unligated forms of the protein. Regarding the structural characteristics of the Rex-detected higher energy state, the correlation between Rex-derived 15N chemical shift differences between the ground and the excited state (|ΔδEG|) as well as the chemical shift difference between apo and holo hI-BABP in the C-terminal half suggests that the higher energy state possesses a more open holo-like conformation in the EFGH-region in the absence of ligands [35]. The connection between thermal melting and the Rex-detected global conformational fluctuations further suggests that the higher energy state carries partially unfolded characteristics with an opening in the EFGH-region [36]. In addition to the abolishment of slow fluctuations below the pKa of H98, changes in |ΔδEG| at pH = 6.3 and above suggest that similarly to bile salt binding, histidine protonation at H98 stabilizes the more open conformation of the EFGH β-strands [39]. Of note, the opening of the EFGH-region is accompanied by a closure of the helical cap and the gap between the helical and the C/D-turn upon both ligand binding [19] and as revealed by MD simulations, in apo hI-BABP as well [36].”
3) Page 7, line211: The analysis of the changes in backbone motions has been performed for the whole proteins, what the authors mean with “the two turn regions”?
Response
By the term “two turn regions” we were referring to the position of the introduced mutations, i.e. that they are located in the vicinity of the C/D- and the G/H-turns. But for better understanding the term was deleted from the sentence.
4) Section 2.3 lines 214-217 and 235-242. Some of the reported experimental details should be moved to Materials and Methods section
Response
Lines 214-217 were moved to Materials and Methods as the reviewer suggested. Lines 235-242 describing the number of residues with Rex and/or τe contribution in the investigated variants we shortened but left in the Results section as we think they are important reporters of mutation induced changes in slow and fast motions.
5) Page 8, lines 248-253: The comparison with CPMG results should be moved in the Discussion section, after presenting relaxation dispersion analysis results
Response
The corresponding lines have been moved to the Discussion as suggested by the reviewer. The sentences were rephrased a bit as follows:
“We note that while in WT and Q99A the position of residues exhibiting conformational exchange according to the model-free analysis (μs-ms timescale) generally fell into the regions exhibiting a nonflat dispersion profile in CPMG relaxation dispersion experiments (0.3 ‑ 10 ms timescale), in Q51A, additional residues were reported by Model-free in the helical cap, βD, and βJ. This indicates that while in WT and Q99A the CPMG Rex-reported global motion is the dominating exchange process, in Q51A μs fluctuations (outside of the CPMG Rex-sensitive timescale) likely to exist.”
6) Page 9, Figure 4. Please specify the criterion to fix a threshold to 0.1. The threshold employed to select ball and stick residues as significant should be represented as a line on the graph. Please label the strands on the 3D structure to increase figure readability.
Response
The threshold of 0.1 was chosen arbitrarily to highlight the differences between the two mutants. It is specified now in the figure caption. The threshold employed for residues in a ball and stick representation has been added to the graph in dotted line. The strands have been labeled to improve readability as the reviewer has suggested.
7) Page 10, line 321. This reviewer believes that the term “slow cluster” is not correct, as it is the exchange process that is slower. Legend of Figure 7 could be rephrased as “residues involved in fast and slow exchange processes”.
Response
The term “slow cluster” was changed to “residues involved in global fluctuation at faster (cluster I) and slower (cluster II) exchange rates” in the caption of Figure 7. The term “slow cluster” was also changed in the main text (line 350 in the revised version) to “cluster II”.
8) Page 16, lines 440-451. The identification of correlated motions is an important issue, however this section of the manuscript is not clear enough. The last paragraph of 2.5 section needs a revision of English language. The clustering method employed to highlight correlated motions should be better described in the results and in the Material and Methods section. A deeper description of Figure 11 is needed to clarify which are the basis of the conclusions drawn from this analysis.
Response
We thank the reviewer to bring this into our attention. We provided further interpretation of the results on the clustering for correlated motions and made some corrections in the Results section and in the Materials and Methods for the better understanding. The temperature of the MD simulations for correlated motions was mistakenly indicated, the correct value is 283 K. Specifically, the following two sentences have been added (pg. 18, lines 499-504 in the revised version):
“Intriguingly, when four clusters were separated, the location of the fourth cluster (green color) showed clear differences between the variants. In the case of WT, it was mainly located around the D strand, while for the mutants, the N-terminal region was involved. In the case of Q99A, the extent of the fourth cluster was significantly larger than for the other variants suggesting an alteration in the separately moving regions.”
9) Page 18-21 Revision of English language is required in some sections. In particular, in the Discussion section, the meaning of some sentences is not clear, some examples:
Line 553 “The strong intrinsic affinity of GCDA for site 1 raises the possibility that it (?) may have a role in the site preference of bile salts in the mixed GCDA/GCA system in hI-BABP”. Please clarify this point.
Response
The sentence has been simplified to: “The strong intrinsic affinity of GCDA for site 1 might have a role in the site preference of bile salts in the mixed GCDA/GCA system in hI-BABP.”
Lines 568-569 “Similar to Q99A, besides the observed chemical shift changes at distant sites, disruption of the hydrophilic network in the vicinity of the C/D-turn by mutation Q51A has long-range dynamic effects as well.”
Response
The sentence has been rephrased to: “Similar to mutation Q99A, besides the structural perturbations at distant sites indicated by chemical shift changes, disruption of the hydrophilic network in the vicinity of the C/D-turn by mutation Q51A has long-range dynamic effects.”
Minor remarks:
1) Figure 1 legend, line 45: “Residues involved in H-bond networks involving...…” instead of “H-bond networks involving…”. Specify that H-bonds are depicted in black
Response
Done.
2) Page 7, line 205 and 206: “in the linker” instead of “a linker”
Response
It has been fixed.
3) Page 8, line 265 “observed” instead of “indicated”
Response
Done.
4) Temperature units should be the same (Kelvin or °C) through the manuscript
Response
Throughout the sections describing the NMR and MD results, we changed the temperature unit to Kelvin in the revised manuscript. In the CD part we would prefer keeping °C as it is used more commonly in folding studies.
5) Page 10, Figure legend 5: “Residues exhibiting a Rex > 15 Hz are marked with a cross”. The cross is not visible
Response
It has been fixed by using a more visible red asterisk.
6) Supplementary Material, Page 1: Bence Balterer name is missing
Response
It has been fixed.
Reviewer 2 Report
The artcile describes the combination of NMR and MD experiments to probe the internal dynamics of hi-BABP protein. Specifically, two residue mutations (Q51A and Q99A) have been exploited to probe the protein motion for the protein in the apo form. The authors used different NMR techniques to investigate protein dynamics for different time ranges.
The article is well written and understandeble; the english level of the manuscript is really good.
Questions:
1) CD spectroscopy has been used to check that mutants do not posses different conformations comparaed to the native form of the system. However, the CD data do not add any interesting information about the protein dynamics for the WT and mutants. Probably this section should be moved in the supporting info.
2) The authors reported that helix melting was different for Q99A compared to WT (line 57-58, Fig.2B), as measured by CD spectroscopy. However, differences seems to be very small and authors should comment on that. Moreover the errors related to sendary structure extimation should be reported as well
3) chemical shift perturbation analysis has been used to check the protein backbone for the WT and mutatns. For Q99A, the authors stated that the region encompassing residue I71-G75 and aminoacid F17 are perturbed in addition to other protein regions. However, any variation in chemical shift fo F17 and I71-G75 are observable for these protein region in fig. 3B. can the authors comment on that?
In general, an analysis of the protein dynamics between the apo- and the holo-forms of the WT and mutats would results more interesting from a physiological point of view.
Author Response
We thank the reviewer for his or her valuable comments and suggestions. Below we list our point-by-point responses:
1) CD spectroscopy has been used to check that mutants do not possess different conformations compared to the native form of the system. However, the CD data do not add any interesting information about the protein dynamics for the WT and mutants. Probably this section should be moved in the supporting info.
Response
We thank the reviewer for his or her comment. We considered presenting the CD results in the Supplementary Information when preparing the manuscript. However, we thought it would be important to show that the mutants are fully folded and must have structures similar to the WT protein. Therefore, if possible, we would prefer keeping part of the CD data in the main text.
2) The authors reported that helix melting was different for Q99A compared to WT (line 57-58, Fig.2B), as measured by CD spectroscopy. However, differences seems to be very small and authors should comment on that. Moreover the errors related to secondary structure estimation should be reported as well.
Response
We thank the reviewer for bringing this into our attention. We added an explanation for the rationale of the experiments and also commented on the error of secondary structure estimation. Specifically, the following sentences have been added to the text:
“The expectable error of the secondary structure estimation is around 5 percent in the absolute values [40, 41], however, because of the similar spectral shape and amplitude of the WT and mutant proteins, it shows more accurately the structural changes in comparisons. These results show that the mutants are fully folded and their overall structure and stability is similar to the WT protein.”
We note that the midpoint temperatures for the unfolding of the helices have been calculated using a two-state unfolding model and added to the results in the revised version (pg. 5, lines 168 and 179). For WT, Q51A, and Q99A they were 53.8 ± 0.7 °C, 53.0 ± 0.8 °C, and 56.3 ± 0.8 °C, respectively.
3) Chemical shift perturbation analysis has been used to check the protein backbone for the WT and mutants. For Q99A, the authors stated that the region encompassing residue I71-G75 and amino acid F17 are perturbed in addition to other protein regions. However, any variation in chemical shift for F17 and I71-G75 are observable for these protein region in fig. 3B. Can the authors comment on that?
Response
Combined (1HN, 15N) chemical shift perturbations at F17 and I71-G75 upon mutation Q99A are slightly exceeding the average. We mention these residues in the text because they are important regions for the subsequent discussion. The words “subtle” and “slightly” have been inserted in the revised version into the corresponding sentence to be more accurate.
-----------------
In general, an analysis of the protein dynamics between the apo- and the holo-forms of the WT and mutants would results more interesting from a physiological point of view.
Response
We agree with the reviewer and a comparative analysis of internal motions in WT and mutated hI-BABPs in homo- and heterotypic complexes with various combinations of di- and trihydroxy bile salts are in progress and will be presented in a subsequent paper.